# Effects of Plasma Temperature in the Blowout Regime for Plasma Accelerators

**Gevy Jiawei Cao** [ID]

Department of Physics, University of Oslo, 0316 Oslo, Norway; jiawei.cao@fys.uio.no

**Abstract:** Research on plasma accelerators for high-energy colliders has rapidly progressed over the past few decades. Plasma acceleration with a high repetition rate will enable higher collider luminosity, but results in a heated plasma. This study investigates two phenomena—beam breakup instability and ion motion—in the nonlinear blowout regime in plasma accelerators and how the plasma temperature affects them. It was found that increasing the plasma temperature enhances the beam breakup instability by reducing the blowout radius while suppressing the on-axis ion-density spike caused by ion motion. This imposes a stringent demand on alignment tolerances, but it offers promising prospects for mitigating ion motion.

**Keywords:** plasma acceleration; high repetition rate; plasma temperature; beam break-up instability (BBU); ion motion; emittance preservation

## 1. Introduction

A major scientific milestone in high-energy physics research occurred in 2012 when CERN announced the discovery of the Higgs boson [1–3]. Precision studies of the Higgs boson may uncover new theories of particles and their interaction mechanisms. In support of continuing the research on Higgs boson, both the European Strategy for Particle Physics [4,5] and the Snowmass process [6] have emphasized the development of an electron–positron collider as a Higgs factory. Starting at a collision energy of around 250 GeV for a Higgs factory, a potential upgrade to a multi-TeV collider is desired to search for new, higher-mass particles. An important objective of such a collider is to reach high luminosity, increasing the number of events per cross-sectional collision area per time.

Advanced accelerator technology can help save the resources required to build a high-energy, high-luminosity electron–positron collider. Traditional accelerators use radio frequency cavities, which provides an acceleration gradient on the order of 10–100 MV/m. Advanced accelerators, such as plasma accelerators [7,8], can provide gradients of 1 GV/m or higher, significantly shortening the length, and therefore the financial resources, required for acceleration.

Research on plasma accelerators has progressed rapidly over the past few years. Major milestones include high-quality, high-efficiency and high-gradient acceleration of electrons in a beam-driven plasma accelerator (PWFA) [9]; the demonstration of free electron lasers using both laser wakefield accelerators (LWFA) [10,11] and PWFA [12]; and the possibility for high-repetition-rate in plasma accelerators [13,14], which leads to higher luminosity. The latter will inevitably result in a high plasma temperature as the beams are injected into the plasma with high frequency, depositing heat along the way.

Some challenges remain in realizing plasma accelerators for high-energy colliders, including beam breakup (BBU) instability [15,16] and ion motion [17], both of which may deteriorate beam quality during acceleration. The BBU instability arises with misaligned bunches: a particle propagating off-axis at any given longitudinal position is affected by the integrated fields in front of it, an effect caused by short-range wakes [18]. The particle will then receive a kick away from the axis, an effect that becomes resonant when combined with

linear focusing. Collectively, a particle bunch will oscillate transversely around the axis with an exponentially increasing amplitude from head to tail and throughout acceleration (known as the beam breakup instability [19]), leading to emittance growth and eventual beam loss.

On the other hand, strong, nonlinear wakefields can be excited in plasma accelerators by an intense, relativistic drive beam [20,21] or an intense laser pulse [22]. A blowout structure is formed as plasma electrons move significantly outward in the radial direction and subsequently rush back inward (attracted by the electrostatic force of uniformly distributed ions), crossing the axis [8,23]. The structure has properties that are desirable when accelerating electrons, such as uniform accelerating fields and linear focusing fields. The linear focusing is provided by exposed plasma ions, which are assumed to be stationary. However, this assumption becomes invalid as the intensity of the accelerated bunch increases. Large ion motion will cause nonlinear focusing fields, thereby disrupting the motion of the accelerated bunch and inducing emittance growth [24], which is undesirable for beam-quality preservation. Therefore, BBU instability has to be avoided and ion motion needs to be mitigated (e.g., through bunch-shaping [25]).

Most studies in plasma acceleration assume cold plasma, which is not the case when the beam repetition rate is on the level of kilohertz or higher. A few studies investigated finite plasma temperature: it has been shown that the amplitude of the peak accelerating field in the blowout regime decreases with increasing plasma temperature [26,27]. Another recent study showed that temperature can linearize focusing fields and reduce emittance growth for positrons [28]. However, beam breakup instability and ion motion have yet to be investigated with finite plasma temperature. A study found that moderate ion motion can mitigate the beam breakup instability, although the high mobility of ions overcompensates for the effects on beam emittance caused by BBU and starts to degrade the beam quality [29]. Therefore, the effects on beam quality from ion motion and BBU instability are not independent.

This work explores the effects of plasma temperature on BBU instability and ion motion, both separately and in combination. It was found that the emittance growth caused by ion motion can be suppressed due to phase mixing of plasma particles as a result of non-zero plasma temperature. On the other hand, beam breakup instability is enhanced because of a reduction in blowout size with finite plasma temperatures.

## 2. Materials and Methods

The study is performed using the particle-in-cell (PIC) code HiPACE++ [30] version 23.05-51. A benchmarking simulation was performed with Snowmass parameters in Table 1 of Ref. [31]. The parameters used for benchmarking are the same as the base parameters used in this study, as shown in Table 1 except for the length of the trailing bunch ($\sigma_z = 20$ μm), the bunch charge (N = $10^{10}$) and the separation distance ($\Delta\xi = 187$ μm). The results show good agreement between HiPACE and QuickPIC simulations, illustrated in Figure 1. Note that beam and plasma densities are sensitive to the resolution of the simulation, which was set to $512 \times 512 \times 256$ for a grid size of $15 \times 15 \times 14.7\ k_p^{-3}$ in both simulations. This is insufficient to resolve the beams, but adequate for the instability study, as the simulation results agree well in Figure 1e. For studies with ion motion, a convergence test was performed and the resolution was increased accordingly to $2048 \times 2048 \times 1024$ for the same grid size.

HiPACE++ is used in this study because the plasma temperature cannot be easily changed in the input file using the open source version of QuickPIC. The beams are initialized with $10^6$ and $10^5$ macroparticles for the drive and trailing bunches, respectively. Plasma particles are simulated with 2 particles per cell. Pre-ionized plasma is used in all the simulations and the beams start immediately in the plasma.

The base parameters used to study the effects of temperature on BBU instability and ion motion are listed in Table 1. These parameters are chosen to provide a bunch charge and bunch length that are collider-relevant and sufficient to observe BBU and ion motion.

A smaller emittance would be required for a collider application, but would require more computation time and resources. The purpose of this study is to investigate the effects of temperature on beam dynamics in plasma accelerators, rather than evaluating how plasma temperatures affect eventual beam parameters for a plasma-based collider. All simulations use an electron driver that excites nonlinear blowout wakefields, followed by an electron trailing bunch. Plasma temperature effects on BBU and ion motion are considered separately first, meaning that ion motion is not modelled in the instability study and the beams are perfectly aligned in the ion motion study. Then, the combined effects of BBU instability, ion motion, and plasma temperature on beam emittance are studied.

**Table 1.** Base plasma and beam parameters for the study of temperature effects on the beam breakup instability and ion motion.

| | |
|---|---|
| **Plasma Density** $[n_o]$ | $2 \times 10^{16}$ cm$^{-3}$ |
| **Initial Energy** $[E_{init}]$ | 25 GeV |
| **Number of Particles, driver** $[N_d]$ | $2.0 \times 10^{10}$ |
| **Number of Particles, trailing** $[N_t]$ | $5.0 \times 10^9$ |
| **Bunch Separation** $[\Delta\xi]$ | 200 μm |
| **Bunch Length, driver** $[\sigma_{z,d}]$ | 40.0 μm |
| **Bunch Length, trailing** $[\sigma_{z,t}]$ | 5.0 μm |
| **Transverse Size, both beams** $[\sigma_r]$ | 0.69 μm |
| **Normalized Emittance, both beams** $[\epsilon_{nr}]$ | 2.0 mm mrad |

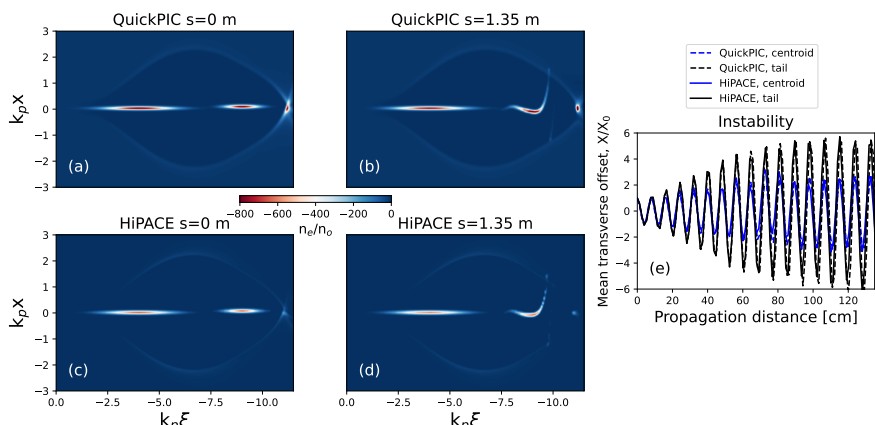

**Figure 1.** QuickPIC and HiPACE simulations benchmarking results. The beams are propagating to the left and the trailing bunch initially has a transverse offset of 3.3 $\sigma_r$. (**a–d**) show the plasma-electron density ($\times 10$ to show the blowout boundary) and beam densities at two timesteps. (**e**) shows the instability growth normalized to the initial offset ($X_0$) for the bunch centroid and the bunch tail (from the center to 2 $\sigma_z$).

## 3. Results

A non-zero plasma temperature provides an initial momentum to plasma electrons—leading to random motion of individual particles—with a net momentum of zero without perturbation. When the plasma is perturbed with a laser or particle beam in the nonlinear blowout regime, the random motion of the plasma electrons reduces the coherence in their subsequent motion. This decoherence of collective motion results in phase mixing and a "smearing" effect on the plasma-electron-density distribution, instead of a sharp, thin blowout sheath followed by an on-axis spike. The effect can be observed in Figure 2a–c.

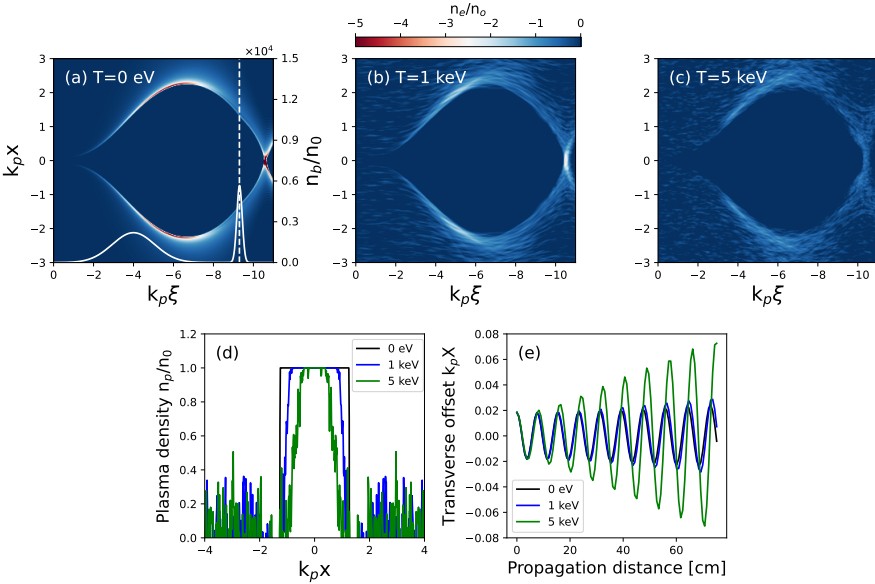

**Figure 2.** Simulations showing (**a**–**c**) the blowout structure at different temperatures, (**d**) the transverse density profile of the plasma ions at the location of the trailing bunch (white dashed line in (**a**)) and (**e**) consequent bunch-tail transverse oscillations with an initial offset of $\sigma_r$. Here, the drive and trailing beam parameters are given in Table 1, with their current profiles shown in (**a**).

### 3.1. Beam Breakup Instability

Beam breakup instability occurs when a particle bunch is transversely misaligned with the wakefields, propagating off-axis. The transverse wake function $W_\perp$ that acts on the beam particles to deflect an initially offset bunch is proportional to $\sim r_b^{-3}(\xi)$, where $r_b$ is the radius of the blowout at the location $\xi$ of the trailing particle. The transverse wake then distorts the bunch over time (see Figure 1b,d); once this occurs, the beam quality is degraded and the beam emittance can no longer be preserved.

With finite plasma temperatures, the size of the blowout is reduced (i.e., smaller $r_b$) as a consequence of the smearing on the plasma-electron density, especially toward the back where the trailing bunch is located. The plasma-ion density profile, equal to the background plasma density $n_0$ inside the blowout, is representative of the blowout radius (shown in Figure 2d). Therefore, it is expected that the beam breakup instability increases with decreasing blowout radius (or increasing plasma temperature). The corresponding transverse offset, with an initial offset of $k_p X = 0.0184$, of the trailing-bunch tail is depicted in Figure 2e, showing an increased transverse-oscillation amplitude with increasing plasma temperature.

As shown in the figure, the effect on BBU is negligible at a temperature $\lesssim 1$ keV, but starts to become problematic at temperatures $\gtrsim 5$ keV. The temperature of a plasma accelerator will increase with the repetition rate and consecutive running time if no heat-extraction technique is applied. The effect will also become larger with longer bunches. Note that energy spread on the trailing bunch can damp BBU instability [32]. The initial energy spread of the trailing bunch is set to zero in all of the simulations. The induced energy spread by the wakefields can differ during beam propagation because there is less energy available for extraction for the trailing bunch at higher temperatures [26,27]. This implies that the increased relative energy spread could damp instabilities at high temperatures. However, as observed in Figure 2e, the instability growth is dominated by the reduction in blowout size at temperatures above $\sim 1$ keV.

### 3.2. Ion Motion

Ion motion is the result of plasma ions reacting to the transverse fields of a high-density electron bunch. The ions are attracted toward the bunch, resulting in an on-axis density

spike. If the electron bunch is sufficiently long, it will observe multiple oscillations of the ions within its length. The amount of ion motion in a nonlinear blowout in cold plasma depends on the bunch charge, bunch length, beam energy and the mass of the plasma ions. Here, lithium plasma is used in the simulations. Because of the reduction in the blowout size, the location of the trailing bunch is adjusted accordingly for the bunch to stay at the back of the blowout (see Figure 3). All the other parameters stay the same, with temperature being the only variable.

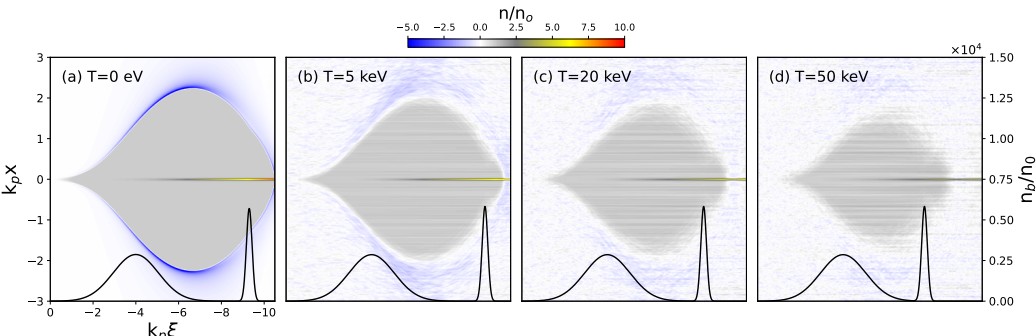

**Figure 3.** Simulations illustrating the on-axis ion densities for the same trailing bunch at different temperatures. Here, the trailing bunch is located at (**a**) $k_p \xi = -9.3$, (**b**) $k_p \xi = -9.3$, (**c**) $k_p \xi = -8.5$, and (**d**) $k_p \xi = -8.0$. All the other parameters are given in Table 1.

Figure 3 shows a decrease in the on-axis ion density with increasing plasma temperature. This is because the same density-smearing effect that happens to electrons also applies to ions. The difference is that the temperature required to observe the effect is much higher for ions, because they have a higher mass and therefore are less mobile than electrons.

Strong ion motion induces nonlinear focusing fields within the accelerated electron bunch (see Figure 4a). A finite plasma temperature spreads out the on-axis ion distribution and linearizes the focusing fields. The consequent emittance evolution is shown in Figure 4b. An et al. [24] found that for cold lithium plasma, the projected beam emittance grows by about 20%, consistent with the 0 eV case here. However, the emittance growth becomes milder with temperature; for example, with $T = 50$ keV, the emittance grows by only $\sim$5%.

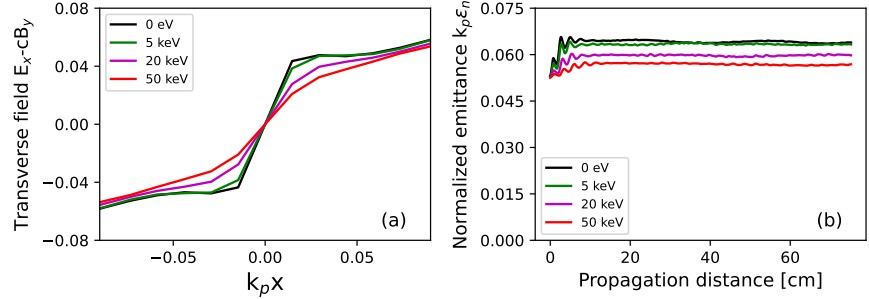

**Figure 4.** The effects of plasma temperature on ion motion: (**a**) the average focusing fields on the accelerated electron bunch, and (**b**) the corresponding emittance evolution.

### 3.3. Combined Effects

From the individual effects on BBU instability and ion motion, we observe that increasing the plasma temperature enhances BBU instability, but reduces the ion density spike. However, ion motion also reduces beam breakup instability by introducing a head-to-tail variation on the focusing fields [29,33]. Therefore, one could imagine an ideal case where moderate ion motion quickly damps BBU instability, but is not sufficient to induce large emittance growth. In order to investigate the combined effects, ion motion is modeled

with an initial transverse beam offset. The resulting transverse oscillations and emittance evolution are shown in Figure 5.

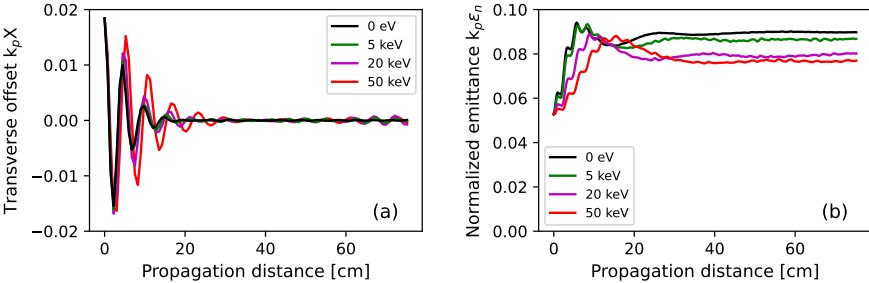

**Figure 5.** The combined effects of ion motion and plasma temperature on beam breakup instability, showing (**a**) the transverse offset of the bunch with an initial offset of $\sigma_r$, and the combined effects on (**b**) the projected beam emittance.

Figure 5a shows that BBU instability can indeed be damped by ion motion (again with an initial offset of $k_p X = 0.0184$). However, the damping happens more slowly when the plasma temperature is higher due to the counteracting effects of temperature on BBU. Moreover, the initial oscillation due to BBU instability causes an emittance growth on the beam, which can be somewhat mitigated by a higher plasma temperature, which smooths the ion density spike and reduces the consequent emittance growth. However, the amount of mitigation is limited (see the difference between $T = 20$ keV and $T = 50$ keV in Figure 5b). This limitation occurs because the initial emittance growth is sufficiently large to almost overshadow the benefits of a more uniform ion distribution.

## 4. Discussion

This study shows that higher plasma temperatures enhance the beam breakup instability but reduce the on-axis density spike caused by ion motion. The two phenomena have opposing effects on emittance growth. When combined, meaning the trailing bunch is initially misaligned and is sufficiently intense to induce significant ion motion, the emittance growth is initially reduced with the increase in temperature as the reduced intensity of the ion density spike dominates. At some temperature, emittance growth caused by BBU instability will cancel the suppression by reduced ion density spike. Eventually, at even higher temperatures, BBU instability will not be sufficiently damped and the beam emittance will continue to grow. Note that the temperature at which this occurs will depend on the beam parameters, the mass of the plasma ions and the initial offset.

Additionally, while not studied here, finite plasma temperatures may reduce the amount of energy available in the wake for extraction (as the plasma electrons are now moving inward incoherently). The effects on energy-transfer efficiency and beam energy spread are subject to further investigation.

In conclusion, the development of high-repetition-rate plasma accelerators, plasma will become increasingly warm. A non-zero plasma temperature sets a strict requirement for alignment tolerances, as increasing temperatures enhance the beam breakup instability. However, an increase in plasma temperature helps to reduce emittance growth by spreading out the ion distribution, effectively reducing the on-axis density spike.

**Funding:** This research received no external funding.

**Data Availability Statement:** All the simulation data are available at: https://uio-my.sharepoint.com/:f:/g/personal/jiaweic_uio_no/ElxMInFlocdLsGtmNmbtWesB6n-HheKy6XGpduZcXGWlhQ?e=7XKhsK, (accessed on 26 October 2023).

**Acknowledgments:** The author would like to thank Jian Bin Ben Chen for useful discussions and for providing part of the benchmarking data. In addition, the author acknowledges Erik Adli, Carl Lindstrøm and Kyrre Sjøbak for their input and discussions on early versions of this work. The computations were performed using resources provided by Sigma2—the National Infrastructure for High Performance Computing and Data Storage in Norway.

**Conflicts of Interest:** The author declares no conflict of interest.

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
