# Peer review of "Effects of Plasma Temperature in the Blowout Regime for Plasma Accelerators"

_instruments, doi:10.3390/instruments7040037_

Round 1

Reviewer 1 Report

Comments and Suggestions for Authors

The manuscript “Effects of Plasma Temperature in the Blowout Regime for Plasma Accelerators”, by Gevy Jiawei Cao, reports numerical studies, using the HiPACE++ PIC code, about the behavior of a particle driven plasma accelerator, tailored on a relevant setting found in the Snowmass document, when the hypothesis of cold plasma is relaxed. The effect of a finite plasma temperature becomes relevant whenever a high repetition rate is considered, since a portion of the energy lost by the driver beam will remain in the plasma, resulting in a temperature increase, whose extent depends on the envisioned heat removal strategy.

The manuscript focuses on how temperature affects two detrimental phenomena, namely the beam breakup instability (BBU) and the consequences of ion motion; the findings are that the temperature induced “smearing” of plasma structures result, from one side, in an enhancement of BBU due to the blowout radius reduction; on the other hand, the same smearing mitigates the focusing force non-linearities due to ion motion. Given the different masses of electrons and ions (a Li gas is assumed in the manuscript) the two effect manifest at different temperature levels, about few keV for electrons and few tens of keV for ions. Since ion motion has the additional effect or reducing BBU instability, it is numerically demonstrated that an optimal working point exists where BBU is (reasonably) suppressed together with the non-linear effects of ion motion, resulting in a mitigated emittance degradation.

The manuscript is well written and easily understood; findings are surely relevant in vision of a high repetition rate / high luminosity plasma based collider and generally well supported by the numerical results. However, before granting publication, I would like the following points to be addressed in order to improve the manuscript:

1) when the optimized working point is analyzed, it is found that a partial emittance decrease occurs. In the text it is described as emittance compensation (line 146) which usually means a re-alignment of bunch longitudinal slices in transverse phase space. However, I think here what get re-aligned are the slices centroids after oscillations due to the initial transverse displacement. I require the Authoress to elaborate on this point and provide a clearer picture of what is happening to the bunch emittance.

2) In the manuscript, the witness energy spread is never mentioned. Since it can have an effect on transverse instabilities, its absence must be justified.

3) In Section 2 it would be advantageous to include some more details, namely: the charge sampling (particle per cell) and whether the plasma is modeled as pre-ionized or not.  How are the bunches initialized? Do beams start inside/outside of plasma?

4) When dealing with BBU, the starting offset value should be included in the main text, not only in the figure captions.

5) Since the paper focuses on particle driven acceleration, it seems weird that the only PWFA driven FEL result published to date [a] is not cited, since LWFA based results are.

[a] R. Pompili et al., Free-electron lasing with compact beam-driven plasma wakefield accelerator, Nature 605, 659 (2022).

Author Response

Dear reviewer, 

Thank you for taking your time to review the article. I have now revised the manuscript based on your comments and suggestions. Here is a detailed response to your specific comments: 

1) I was not aware of the commonly-known meaning of "emittance compensation". What I wanted to say is simply that the temperature effects on ion motion mitigate emittance growth, while as the effects on BBU enhance it --- and these two effects work against each other, providing partial mitigation for emittance growth. The wording has been changed from "compensation" to "mitigation". 

2) This is a very good point. The initial energy spread of the trailing bunch is 0. The induced energy spread during acceleration is around 1% in the cold plasma case and higher in warm plasma. The higher energy spread in warm plasma is a direct result of plasma temperature --- because at higher temperatures there is less energy available for extraction, as shown by earlier papers studying temperature effects on the peak of the accelerating field. This can certainly have an effect on BBU, and is partially reflected by the negligible difference between T=0keV and T=1keV case in the BBU study. A few sentences have been added to reflect the potential effect coming from energy spread. However, as higher temperatures are reached, the effects are dominated by the reduction in the size of the blowout. 

3) More details have been added as requested. 

4) The offset is now stated in both the main text and the figure captions. 

5) The suggested reference is now added to the introduction section. 

Reviewer 2 Report

Comments and Suggestions for Authors

The paper is sound and is complementary to previous works realized on the effect of temperature on the peak accelerating electric field. The range of parameters, in particular the accumulative thermal effect, is interesting. The choice of the parameters presented in Table 1 should be explained in a clear way since they are crucial on the effect under study. It should be also important for the reader to see the parameters used (reference 30) for the benchmark of the actual simulation in this paper and to understand also why the QuickPIC code cannot be used for assessing the effect of plasma temperature.

The paper can be published after these minor suggested corrections which  can easily be introduce in this paper. 

Author Response

Dear reviewer, 

Thank you for taking your time to review this article. I have now revised the article based on your comments and suggestions. Here is a detailed response to your specific comments: 

  1. Two sentences have been added to justify the choice of parameters in Table 1: These parameters are chosen to provide a bunch charge and bunch length that are collider relevant and sufficient to observe BBU and ion motion. A smaller emittance would be required for a collider application, but would require more computation time and resources. The purpose of this study is to investigate the effects of temperature on beam dynamics in plasma accelerators, rather than evaluating how plasma temperatures affect eventual beam parameters for a plasma-based collider.
  2. The parameters used for benchmarking are very similar to the ones listed in Table 1. The quantitative differences (double the charge, 4 times longer bunch length and different separation distance for correct loading) are now stated in the main text. 
  3. The open source version of QuickPIC does not provide the possibility to change the plasma temperature through the input file. In addition, HiPACE++ is GPU-compatible, making it a more efficient code than QuickPIC which can only run on CPUs. This has now been stated in the text.